# Predict Responsibly: Improving Fairness and Accuracy by Learning to Defer

**David Madras, Toniann Pitassi & Richard Zemel**
University of Toronto
Vector Institute
{madras,toni,zemel}@cs.toronto.edu

## Abstract

In many machine learning applications, there are multiple decision-makers involved, both automated and human. The interaction between these agents often goes unaddressed in algorithmic development. In this work, we explore a simple version of this interaction with a two-stage framework containing an automated model and an external decision-maker. The model can choose to say PASS, and pass the decision downstream, as explored in rejection learning. We extend this concept by proposing *learning to defer*, which generalizes rejection learning by considering the effect of other agents in the decision-making process. We propose a learning algorithm which accounts for potential biases held by external decision-makers in a system. Experiments demonstrate that learning to defer can make systems not only more accurate but also less biased. Even when working with inconsistent or biased users, we show that deferring models still greatly improve the accuracy and/or fairness of the entire system.

## 1 Introduction

Can humans and machines make decisions jointly? A growing use of automated decision-making in complex domains such as loan approvals [5], medical diagnosis [14], and criminal justice [26], has raised questions about the role of machine learning in high-stakes decision-making, and the role of humans in overseeing and applying machine predictions. Consider a black-box model which outputs risk scores to assist a judge presiding over a bail case [19]. How does a risk score factor into the decision-making process of an external agent such as a judge? How should this influence how the score is learned? The model producing the score may be state-of-the-art in isolation, but its true impact comes as an element of the judge's decision-making process.

We argue that since these models are often used as part of larger systems e.g. in tandem with another decision maker (like a judge), they should learn to predict *responsibly*: the model should predict only if its predictions are reliably aligned with the system's objectives, which often include accuracy (predictions should mostly indicate ground truth) and fairness (predictions should be unbiased with respect to different subgroups).

Rejection learning [8, 10] proposes a solution: allow models to *reject* (not make a prediction) when they are not confidently accurate. However, this approach is inherently *nonadaptive*: both the model and the decision-maker act independently of one another. When a model is working in tandem with some external decision-maker, the decision to reject should depend not only on the model's confidence, but also on the decision-maker's expertise and weaknesses. For example, if the judge's black-box is uncertain about some subgroup, but the judge is very inaccurate or biased towards that subgroup, we may prefer the model make a prediction despite its uncertainty.

Our main contribution is the formulation of adaptive rejection learning, which we call *learning to defer*, where the model works *adaptively* with the decision-maker. We provide theoretical and experimental evidence that learning to defer improves upon the standard rejection learning paradigm, if models are intended to work as part of a larger system. We show that embedding a deferring model in a pipeline can improve the accuracy and fairness of the system as a whole. Experimentally, we simulate three scenarios where our model can defer judgment to external decision makers, echoing realistic situations where downstream decision makers are inconsistent, biased, or have access to side information. Our experimental results show that in each scenario, learning to defer allows models to work with users to make fairer, more responsible decisions.

## 2 Learning to Defer

### 2.1 A Joint Decision-Making Framework

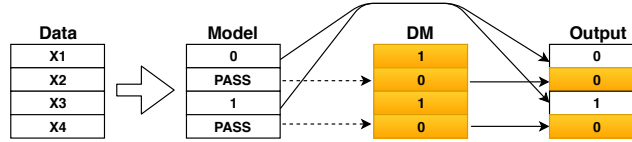

Figure 1: A larger decision system containing an automated model. When the model predicts, the system outputs the model's prediction; when the model says PASS, the system outputs the decision-maker's (DM's) prediction. Standard rejection learning considers the model stage, in isolation, as the system output, while learning-to-defer optimizes the model over the system output.

A complex real-world decision system can be modeled as an interactive process between various agents including decision makers, enforcement groups, and learning systems. Our focus in this paper is on a two-agent model, between one decision-maker and one learning model, where the decision flow is in two stages. This simple but still interactive setup describes many practical systems containing multiple decision-makers (Fig 1). The first stage is an automated model whose parameters we want to learn. The second stage is some external decision maker (DM) which we do not have control over e.g. a human user, a proprietary black-box model. The decision-making flow is modeled as a cascade, where the first-step model can either predict (positive/negative) or say PASS. If it predicts, the DM will output the model's prediction. However, if it says PASS, the DM makes its own decision. This scenario is one possible characterization of a realistic decision task, which can be an interactive (potentially game-theoretic) process.

We can consider the first stage to be flagging difficult cases for review, culling a large pool of inputs, auditing the DM for problematic output, or simply as an assistive tool. In our setup, we assume that the DM has access to information that the model does not — reflecting a number of practical scenarios where DMs later in the chain may have more resources for efficiency, security, or contextual reasons. However, the DM may be flawed, e.g. biased or inconsistent. A tradeoff suggests itself: can a machine learning model be combined with the DM to leverage the DM's extra insight, but overcome its potential flaws?

We can describe the problem of learning an automated model in this framework as follows. There exist data $X \in \mathbb{R}^n$, ground truth labels $Y \in \{0, 1\}$, and some auxiliary data $Z \in \mathbb{R}^m$ which is only available to the DM. If we let $s \in \{0, 1\}$ be a PASS indicator variable ($s = 1$ means PASS), then the joint probability of the system in Fig. 1 can be expressed as follows:

$$P_{defer}(Y|X, Z) = \prod_i [P_M(Y_i = 1|X_i)^{Y_i}(1 - P_M(Y_i = 1|X_i))^{1-Y_i}]^{(1-s_i|X_i)}$$

$$[P_D(Y_i = 1|X_i, Z_i)^{Y_i}(1 - P_D(Y_i = 1|X_i, Z_i))^{1-Y_i}]^{(s_i|X_i)}$$

(1)

where $P_M$ is the probability assigned by the automated model, $P_D$ is the probability assigned by the DM, and $i$ indexes examples. This can be seen as a mixture of Bernoullis, where the labels are generated by either the model or the DM as determined by $s$. For convenience, we compress the probabilistic notation:

$$\hat{Y}_M = f(X) = P_M(Y = 1|X) \in [0, 1]; \quad \hat{Y}_D = h(X, Z) = P_D(Y = 1|X, Z) \in [0, 1]$$

$$\hat{Y} = (1 - s)\hat{Y}_M + s\hat{Y}_D \in [0, 1]; \quad s = g(X) \in \{0, 1\}$$

(2)

$\hat{Y}_M, \hat{Y}_D, \hat{Y}$ are model predictions, DM predictions, and system predictions, respectively (left to right in Fig. 1). The DM function $h$ is a fixed, unknown black-box. Therefore, learning good $\{\hat{Y}_M, s\}$ involves learning functions $f$ and $g$ which can adapt to $h$ – the goal is to make $\hat{Y}$ a good predictor of $Y$. To do so, we want to find the maximum likelihood solution of Eq. 1. We can minimize its negative log-likelihood $\mathcal{L}_{defer}$, which can be written as:

$$\mathcal{L}_{defer}(Y, \hat{Y}_M, \hat{Y}_D, s) = -\log P_{defer}(Y|X, Z) = -\sum_i [(1 - s_i)\ell(Y_i, \hat{Y}_{M,i}) + s_i \ell(Y_i, \hat{Y}_{D,i})] \quad (3)$$

where $\ell(Y, p) = Y \log p + (1 - Y) \log(1 - p)$ i.e. the log probability of the label with respect to some prediction $p$. Minimizing $\mathcal{L}_{defer}$ is what we call ***learning to defer***. In learning to defer, we aim to learn a model which outputs predictive probabilities $\hat{Y}_M$ and binary deferral decisions $s$, in order to optimize the output of the *system as a whole*. The role of $s$ is key here: rather than just an expression of uncertainty, we can think of it as a gating variable, which tries to predict whether $\hat{Y}_M$ or $\hat{Y}_D$ will have lower loss on any given example. This leads naturally to a mixture-of-experts learning setup; however, we are only able to optimize the parameters for one expert ($\hat{Y}_M$), whereas the other expert ($\hat{Y}_D$) is out of our control. We discuss further in Sec. 3.

We now examine the relationship between learning to defer and rejection learning. Specifically, we will show that learning to defer is a generalization of rejection learning and argue why it is an important improvement over rejection learning for many machine learning applications.

## 2.2 Learning to Reject

Rejection learning is the predominant paradigm for learning models with a PASS option (see Sec. 4). In this area, the standard method is to optimize the accuracy-rejection tradeoff: how much can a model improve its accuracy on the cases it *does* classify by PASS-ing some cases? This is usually learned by minimizing a classification objective $\mathcal{L}_{reject}$ with a penalty $\gamma_{reject}$ for each rejection [10], where $Y$ is ground truth, $\hat{Y}_M$ is the model output, and $s$ is the reject variable ($s = 1$ means PASS); all binary:

$$\mathcal{L}_{reject}(Y, \hat{Y}_M, s) = -\sum_i [(1 - s_i)\ell(Y_i, \hat{Y}_{M,i}) + s_i \gamma_{reject}] \quad (4)$$

where $\ell$ is usually classification accuracy, i.e. $\ell(Y_i, \hat{Y}_i) = \mathbb{1}[Y_i = \hat{Y}_i]$. If we instead consider $\ell$ to be the log-probability of the label, then we can interpret $\mathcal{L}_{reject}$ probabilistically as the negative log-likelihood of the joint distribution $P_{reject}$:

$$P_{reject}(Y|X) = \prod_i [\hat{Y}_{M,i}^{Y_i}(1 - \hat{Y}_{M,i})^{(1-Y_i)}]^{1-s_i} \exp(\gamma_{reject})^{s_i} \quad (5)$$

## 2.3 Learning to Defer is Adaptive Rejection Learning

In learning to defer, the model leverages information about the DM to make PASS decisions adaptively. We can consider how learning to defer relates to rejection learning. Examining their loss functions Eq. 3 and Eq. 4 respectively, the only difference is that the rejection loss has a constant $\gamma_{reject}$ where the deferring loss has variable $\ell(Y, \hat{Y}_D)$. This leads us to the following:

**Theorem.** *Let $\ell(Y, \hat{Y})$ be our desired example-wise objective, where $Y = \arg\min_{\hat{Y}} -\ell(Y, \hat{Y})$. Then, if the DM has constant loss (e.g. is an oracle), there exist values of $\gamma_{reject}, \gamma_{defer}$ for which the learning-to-defer and rejection learning objectives are equivalent.*

**Proof.** As in Eq. 4, the standard rejection learning objective is

$$\mathcal{L}_{reject}(Y, \hat{Y}_M, \hat{Y}_D, s) = -\sum_i [(1 - s_i)\ell(Y_i, \hat{Y}_{M,i}) + s_i \gamma_{reject}] \quad (6)$$

where the first term encourages a low negative loss $\ell$ for non-PASS examples and the second term penalizes PASS at a constant rate, $\gamma_{reject}$. If we include a similar $\gamma_{defer}$ penalty, the deferring loss function is

$$\mathcal{L}_{defer}(Y, \hat{Y}_M, \hat{Y}_D, s) = -\sum_i [(1 - s_i)\ell(Y_i, \hat{Y}_{M,i}) + s_i \ell(Y_i, \hat{Y}_{D,i}) + s_i \gamma_{defer}] \quad (7)$$

Now, if the DM has constant loss, meaning $\ell(Y, \hat{Y}_D) = \alpha$, we have (with $\gamma_{defer} = \gamma_{reject} - \alpha$):

$$
\begin{aligned}
\mathcal{L}_{defer}(Y, \hat{Y}_M, \hat{Y}_D, s) &= -\sum_i [(1 - s_i)\ell(Y_i, \hat{Y}_{M,i}) + s_i \cdot \alpha + s_i \gamma_{defer}] \\
&= -\sum_i [(1 - s_i)\ell(Y_i, \hat{Y}_{M,i}) + s_i(\gamma_{defer} + \alpha)] = \mathcal{L}_{reject}(Y, \hat{Y}_{M,i}, s)
\end{aligned}
\tag{8}
$$

∎

## 2.4 Why Learn to Defer?

The proof in Sec. 2.3 shows the central point of learning to defer: rejection learning is exactly a special case of learning to defer: a DM with constant loss $\alpha$ on each example. We find that the adaptive version (learning to defer), more accurately describes real-world decision-making processes. Often, a PASS is not the end of a decision-making sequence. Rather, a decision must be made eventually on every example by a DM, whether the automated model predicts or not, and the DM will not, in general, have constant loss on each example.

Say our model is trained to detect melanoma, and when it says PASS, a human doctor can run an extra suite of medical tests. The model learns that it is very inaccurate at detecting amelanocytic (non-pigmented) melanoma, and says PASS if this might be the case. However, suppose that the doctor is even *less* accurate at detecting amelanocytic melanoma than the model is. Then, we may prefer the model to make a prediction despite its uncertainty. Conversely, if there are some illnesses that the doctor knows well, then the doctor may have a more informed, nuanced opinion than the model. Then, we may prefer the model say PASS more frequently relative to its internal uncertainty.

Saying PASS on the wrong examples can also have fairness consequences. If the doctor's decisions bias against a certain group, then it is probably preferable for a less-biased model to defer less frequently on the cases of that group. If some side information helps a DM achieve high accuracy on some subgroup, but confuses the DM on another, then the model should defer most frequently on the DM's high accuracy subgroup, to ensure fair and equal treatment is provided to all groups. In short, if the model we train is part of a larger pipeline, then we should train and evaluate the performance of *the pipeline with this model included*, rather than solely focusing on the model itself. We note that it is unnecessary to acquire decision data from a specific DM; rather, data could be sampled from many DMs (potentially using crowd-sourcing). Research suggests that common trends exist in DM behavior [6, 11], suggesting that a model trained on some DM or group of DMs could generalize to unseen DMs.

## 3 Formulating Adaptive Models within Decision Systems

In our decision-making pipeline, we aim to formulate a fair model which can be used for learning to defer (Eq. 3) (and by extension non-adaptive rejection learning as well (Eq. 4)). Such a model must have two outputs for each example: a predictive probability $\hat{Y}_M$ and a PASS indicator $s$. We draw inspiration from the mixture-of-experts model [21]. One important difference between learning-to-defer and a mixture-of-experts is that one of the "experts" in this case is the DM, which is out of our control; we can only learn the parameters of $\hat{Y}_M$.

If we interpret the full system as a mixture between the model's prediction $\hat{Y}_M$ and the DM's predictions $\hat{Y}_D$, we can introduce a mixing coefficient $\pi$, where $s \sim Ber(\pi)$. $\pi$ is the probability of deferral, i.e. that the DM makes the final decision on an example $X$, rather than the model; $1 - \pi$ is the probability that the model's decision becomes the final output of the system. Recall that $\hat{Y}_M, \pi$ are functions of the input $X$; they are parametrized below by $\theta$. Then, if there is some loss $\ell(Y, \hat{Y})$ we want our system to minimize, we can learn to defer by minimizing an expectation over $s$:

$$
\begin{aligned}
\mathcal{L}_{defer}(Y, \hat{Y}_M, \hat{Y}_D, \pi; \theta) &= \mathbb{E}_{s \sim Ber(\pi)} \mathcal{L}(Y, \hat{Y}_M, \hat{Y}_D, s; \theta) \\
&= \sum_i \mathbb{E}_{s_i \sim Ber(\pi_i)}[(1 - s_i)\ell(Y_i, \hat{Y}_{M,i}; \theta) + s_i \ell(Y_i, \hat{Y}_{D,i})]
\end{aligned}
\tag{9}
$$

or, in the case of rejection learning:

$$\mathcal{L}_{reject}(Y, \hat{Y}_M, \hat{Y}_D, \pi; \theta) = \sum_i \mathbb{E}_{s_i \sim Ber(\pi_i)}[(1 - s_i)\ell(Y_i, \hat{Y}_{M,i}; \theta) + s_i \gamma_{reject}] \qquad (10)$$

Next, we give two methods of specifying and training such a model and present our method of learning these models fairly, using a regularization scheme.

### 3.1 Post-hoc Thresholding

One way to formulate an adaptive model with a PASS option is to let $\pi$ be a function of $\hat{Y}_M$ alone; i.e. $\hat{Y}_M = f(X)$ and $\pi = g(\hat{Y}_M)$. One such function $g$ is a thresholding function — we can learn two thresholds $t_0, t_1$ (see Figure 2) which yield a ternary classifier. The third category is PASS, which can be outputted when the model prefers not to commit to a positive or negative prediction. A convenience of this method is that the thresholds can be trained post-hoc on an existing model with an output in $[0, 1]$ e.g. many binary classifiers. We use a neural network as our binary classifier, and describe our post-hoc thresholding scheme in Appendix D. At test time, we use the thresholds to partition the examples. On each example, the model outputs a score $\beta \in [0, 1]$. If $t_0 < \beta < t_1$, then we

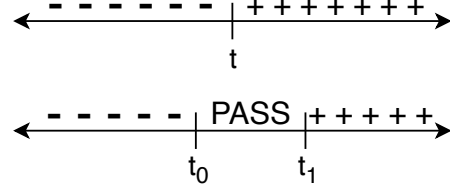

Figure 2: Binary classification (one threshold) vs. ternary classification with a PASS option (two thresholds)

output $\pi = 1$ and defer (the value of $\hat{Y}_M$ becomes irrelevant). Otherwise, if $t_0 \geq \beta$ we output $\pi = 0, \hat{Y}_M = 0$; if $t_1 \leq \beta$ we output $\pi = 0, \hat{Y}_M = 1$. Since $\pi \in \{0, 1\}$ here, the expectation over $s \sim Ber(\pi)$ in Eq. 9 is trivial.

### 3.2 Learning a Differentiable Model

We may wish to use continuous outputs $\hat{Y}_M, \pi \in [0, 1]$ and train our models with gradient-based optimization. To this end, we consider a method for training a differentiable adaptive model. One could imagine extending the method in Sec. 3.1 to learn smooth thresholds end-to-end on top of a predictor. However, to add flexibility, we can allow $\pi$ to be a function of $X$ as well as $\hat{Y}_M$, i.e. $\hat{Y}_M = f(X)$ and $\pi = g(\hat{Y}_M, X)$. This is advantageous because a DM's actions may depend heterogenously on the data: the DM's expected loss may change as a function of $X$, and it may do so differently than the model's. We can parametrize $\hat{Y}_M$ and $\pi$ with neural networks, and optimize Eq. 9 or 10 directly using gradient descent. At test time, we defer when $\pi > 0.5$.

We estimate the expected value in Eq. 9 by sampling $s \sim Ber(\pi)$ during training (once per example). To estimate the gradient through this sampling procedure, we use the Concrete relaxation [28]. Additionally, it can be helpful to stop the gradient from $\pi$ from backpropagating through $\hat{Y}_M$. This allows for $\hat{Y}_M$ to still be a good predictor independently of $\pi$.

See Appendix F for a brief discussion of a third model we consider, a Bayesian Neural Network [3].

### 3.3 Fair Classification through Regularization

We can build a regularized fair loss function to combine error rate with a fairness metric. We can extend the loss in Eq. 9 to include a regularization term $\mathcal{R}$, with a coefficient $\alpha_{fair}$ to balance accuracy and fairness:

$$\mathcal{L}_{defer}(Y, \hat{Y}_M, \hat{Y}_D, \pi; \theta) = \mathbb{E}_{s \sim Ber(\pi)}[\mathcal{L}(Y, \hat{Y}_M, \hat{Y}_D, s; \theta) + \alpha_{fair}\mathcal{R}(Y, \hat{Y}_M, \hat{Y}_D, s) \qquad (11)$$

We now provide some fairness background. In fair binary classification, we have input labels $Y$, predictions $\hat{Y}$, and sensitive attribute $A$ (e.g., gender, race, age, etc.), assuming for simplicity that $Y, \hat{Y}, A \in \{0, 1\}$. In this work we assume that $A$ is known. The aim is twofold: firstly, *accurate*

classification i.e $Y_i = \hat{Y}_i$; and *fairness with respect to A* i.e. $\hat{Y}$ does not discriminate unfairly against particular values of $A$. Adding fairness constraints can provably hurt classification error [29]. We thus define a loss function which trades off between these two objectives, yielding a regularizer. We choose equalized odds as our fairness metric [20], which requires that false positive and false negative rates are equal between the two groups. We will refer to the difference between these rates as disparate impact (DI). Here we define a continuous relaxation of DI, having the model output a probability $p$ and considering $\hat{Y} \sim Ber(p)$. The resulting term $\mathscr{D}$ acts as our regularizer $\mathcal{R}$ in Eq. 11:

$$DI_{Y=i}(Y, A, \hat{Y}) = |\mathbb{E}_{\hat{Y} \sim Ber(p)}(\hat{Y} = 1 - Y | A = 0, Y = i) - \mathbb{E}_{\hat{Y} \sim Ber(p)}(\hat{Y} = 1 - Y | A = 1, Y = i)|$$

$$\mathscr{D}(Y, A, \hat{Y}) = \frac{1}{2}(DI_{Y=0}(Y, A, \hat{Y}) + DI_{Y=1}(Y, A, \hat{Y}))$$

$$(12)$$

Our regularization scheme is similar to Bechavod and Ligett [2], Kamishima et al. [24]; see Appendix B for results confirming the efficacy of this scheme in binary classification. We show experimentally that the equivalence between learning to defer with an oracle-DM and rejection learning holds in the fairness case (see Appendix E).

## 4 Related Work

**Notions of Fairness.** A challenging aspect of machine learning approaches to fairness is formulating an operational definition. Several works have focused on the goal of treating similar people similarly (individual fairness) and the necessity of fair-awareness [13, 35].

Some definitions of fairness center around statistical parity [23, 24], calibration [17, 30] or equalized odds [7, 20, 27, 34]. It has been shown that equalized odds and calibration cannot be simultaneously (non-trivially) satisfied [7, 27]. Hardt et al. [20] present the related notion of "equal opportunity". Zafar et al. [34] and Bechavod and Ligett [2] develop and implement learning algorithms that integrate equalized odds into learning via regularization.

**Incorporating PASS.** Several works have examined the PASS option (cf. *rejection learning*), beginning with Chow [8, 9] who studies the tradeoff between error-rate and rejection rate. Cortes et al. [10] develop a framework for integrating PASS directly into learning. Attenberg et al. [1] discuss the difficulty of a model learning what it doesn't know (particularly rare cases), and analyzes how human users can audit such models. Wang et al. [33] propose a cascading model, which can be learned end-to-end. However, none of these works look at the fairness impact of this procedure. From the AI safety literature, Hadfield-Menell et al. [18] give a reinforcement-learning algorithm for machines to work with humans to achieve common goals. We also note that the phrase "adaptive rejection" exists independently of this work, but with a different meaning [15].

A few papers have addressed topics related to both above sections. Bower et al. [4] describe fair sequential decision making but do not have a PASS concept or provide a learning procedure. In Joseph et al. [22], the authors show theoretical connections between KWIK-learning and a proposed method for fair bandit learning. Grgić-Hlaca et al. [16] discuss fairness that can arise out of a mixture of classifiers. Varshney and Alemzadeh [32] propose "safety reserves" and "safe fail" options which combine learning with rejection and fairness/safety, but do not analyze learning procedures or larger decision-making frameworks. [31] argue that the choice to model "only certain technical parts of the system" is a flaw in many approaches to fair ML; our method is a step towards addressing what this paper calls "the framing trap".

## 5 Experiments

We experiment with three scenarios, each of which represent an important class of real-world decision-makers:

1. **High-accuracy DM**, ignores fairness: This may occur if the extra information available to the DM is important, yet withheld from the model for privacy or computational reasons.
2. **Highly-biased DM**, strongly unfair: Particularly in high-stakes/sensitive scenarios, DMs can exhibit many biases.

3. **Inconsistent DM**, ignores fairness (DM's accuracy varies by subgroup, with total accuracy lower than model): Human DMs can be less accurate, despite having extra information [12]. We add noise to the DM's output on some subgroups to simulate human inconsistency.

Due to difficulty obtaining and evaluating real-life decision-making data, we use "semi-synthetic data": real datasets, and simulated DM data by training a separate classifier under slightly different conditions (see Experiment Details). In each scenario, the simulated DM receives access to extra information which the model does not see.

**Datasets and Experiment Details**. We use two datasets: COMPAS [26], where we predict a defendant's recidivism without discriminating by race, and Heritage Health (https://www.kaggle.com/c/hhp), where we predict a patient's Charlson Index (a comorbidity indicator) without discriminating by age. We train all models and DMs with a fully-connected two-layer neural network [1]. See Appendix C for details on datasets and experiments.

We found post-hoc and end-to-end models performed qualitatively similarly for high-accuracy DMs, so we show results from the simpler model (post-hoc) for those. However, the post-hoc model cannot adjust to the case of the inconsistent DM (scenario 3), since it does not take $X$ as an input to $\pi$ (as discussed in Sec. 3.2), so we show results from the end-to-end model for the inconsistent DM.

Each DM receives extra information in training. For COMPAS, this is the defendant's violent recidivism; for Health, this is the patient's primary condition group. To simulate high-bias DMs (scen. 2) we train a regularized model with $\alpha_{fair} = -0.1$ to encourage learning a "DM" with *high* disparate impact. To create inconsistent DMs (scen. 3), we flip a subset of the DM's predictions post-hoc with 30% probability: on COMPAS, this subset is people below the mean age; on Health this is males.

**Displaying Results**. We show results across various hyperparameter settings ($\alpha_{fair}, \gamma_{defer}/\gamma_{reject}$), to illustrate accuracy and/or fairness tradeoffs. Each plotted line connects several points, which are each a median of 5 runs at one setting. In Fig. 6, we only show the Pareto front, i.e., points for which no other point had both better accuracy and fairness. All results are on held-out test sets.

## 5.1 Learning to Defer to Three Types of DM

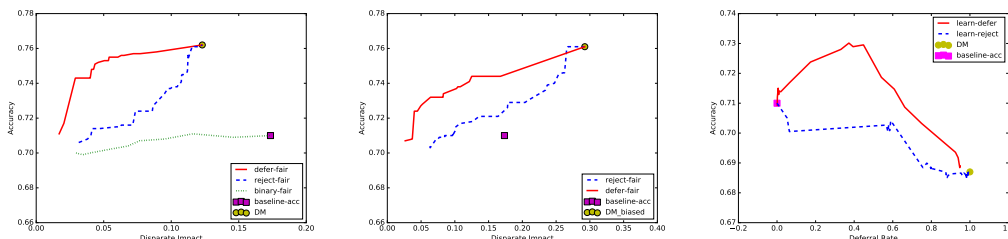

(a) COMPAS, High-Accuracy DM    (b) COMPAS, Highly-Biased DM    (c) COMPAS, Inconsistent DM

Figure 3: Comparing learning-to-defer, rejection learning and binary models. dataset only; Health dataset results in Appendix A. Each figure is a different DM scenario. In Figs. 3a and 3b, X-axis is fairness (lower is better); in Fig. 3c, X-axis is deferral rate. Y-axis is accuracy for all figures. Square is a baseline binary classifier, trained only to optimize accuracy; dashed line is fair rejection model; solid line is fair deferring model. Yellow circle is DM alone. In Fig. 3a, green dotted line is a binary model also optimizing fairness. Figs. 3a and 3b are hyperparameter sweep over $\gamma_{reject/defer}/\alpha_{fair}$; Fig. 3c sweeps $\gamma_{reject/defer}$ only, with $\alpha_{fair} = 0$ (for $\alpha_{fair} \geq 0$, see Appendix G).

**High-Accuracy DM**. In this experiment, we consider the scenario where a DM has higher accuracy than the model we train, due to the DM having access to extra information/resources for security, efficiency, or contextual reasons. However, the DM is not trained to be fair. In Fig. 3a, we show that learning-to-defer achieves a better accuracy-fairness tradeoff than rejection learning. Hence, learning-to-defer can be a valuable fairness tool for anyone who designs or oversees a many-part system - an adaptive first stage can improve the fairness of a more accurate DM. The fair rejection learning model also outperforms binary baselines, by integrating the extra DM accuracy on some

examples. For further analysis, we break out the results in Figs. 3a by deferral rate, and find that most of the benefit in this scenario is indeed coming from added fairness by the model (see Appendix H).

**Highly-Biased DM**. In this scenario, we consider the case of a DM which is extremely biased (Fig. 3b). We find that the advantage of a deferring model holds in this case, as it adapts to the DM's extreme bias. For further analysis, we examine the deferral rate of each model in this plot (see Appendix I). We find that the deferring model's adaptivity brings two advantages: it can adaptively defer at different rates for the two sensitive groups to counteract the DM's bias; and it is able to modulate the overall amount that it defers when the DM is biased.

**Inconsistent DM**. In this experiment, we consider a DM with access to extra information, but which due to inconsistent accuracy across subgroups, has a lower overall accuracy than the model. In Fig. 3c, we compare deferring and rejecting models, examining their classification accuracy at different deferral rates. We observe that for each deferral rate, the model that learned to defer achieves a higher classification accuracy. Furthermore, we find that the best learning-to-defer models outperform both the DM and a baseline binary classifier. Note that although the DM is less accurate than the model, the most accurate result is not to replace the DM, but to use a DM-model mixture. Critically, only when the model is adaptive (i.e. learns to defer) is the potential of this mixture unlocked.

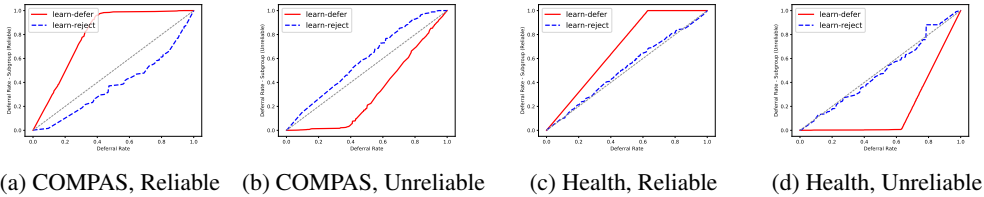

| (a) COMPAS, Reliable | (b) COMPAS, Unreliable | (c) Health, Reliable | (d) Health, Unreliable |

Figure 4: Each point is some setting of $\gamma_{reject/defer}$. X-axis is total deferral rate, Y-axis is deferral rate on DM reliable/unreliable subgroup (COMPAS: Old/Young; Health: Female/Male). Gray line $= 45°$: above is more deferral; below is less. Solid line: learning to defer; dashed line: rejection learning.

To analyze further how the deferring model in Fig. 3c achieves its accuracy, we examine two subgroups from the data: where the DM is reliable and unreliable (the unreliable subgroup is where post-hoc noise was added to the DM's output; see Experiment Details). Fig. 4 plots the deferral rate on these subgroups against the overall deferral rates. We find that the deferring models deferred more on the DM's reliable subgroup, and less on the unreliable subgroup, particularly as compared to rejection models. This shows the advantage of learning to defer; the model was able to adapt to the strengths and weaknesses of the DM.

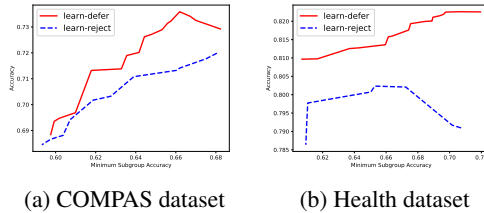

| (a) COMPAS dataset | (b) Health dataset |

Figure 5: Each point is a single run in sweep over $\gamma_{reject}/\gamma_{defer}$. X-axis is the model's lowest accuracy over 4 subgroups, defined by the cross product of binarized (sensitive attribute, unreliable attribute), which are (race, age) and (age, gender) for COMPAS and Health respectively. Y-axis is model accuracy. Only best Y-value for each X-value shown. Solid line is learning to defer; dashed line is rejection learning.

We also explore how learning-to-defer's errors distribute across subgroups. We look at accuracy on four subgroups, defined by the cross-product of the sensitive attribute and the attribute defining the DM's unreliability, both binary. In Fig. 5, we plot the minimum subgroup accuracy (MSA) and the overall accuracy. We find that the deferring models (which were higher accuracy in general), continue to achieve higher accuracy even when requiring that models attain a certain MSA. This means that the improvement we see in the deferring models are not coming at the expense of the least accurate subgroups. Instead, we find that the most accurate deferring models also have the highest MSA, rather than exhibiting a tradeoff. This is a compelling natural fairness property of learning to defer which we leave to future work for further investigation.

## 6    Conclusion

In this work, we define a framework for multi-agent decision-making which describes many practical systems. We propose a method, learning to defer (or adaptive rejection learning), which generalizes rejection learning under this framework. We give an algorithm for learning to defer in the context of larger systems and explain how to do so fairly. Experimentally, we demonstrate that deferring models can optimize the performance of decision-making pipelines as a whole, beyond the improvement provided by rejection learning. This is a powerful, general framework, with ramifications for many complex domains where automated models interact with other decision-making agents. Through deferring, we show how models can learn to predict responsibly within their surrounding systems, an essential step towards fairer, more responsible machine learning.

## Footnotes

[1]Code available at `https://github.com/dmadras/predict-responsibly`.

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
