[Supplementary Material]

# A    Learning to Defer to Three Types of DM: Health Results

(a) Health, High-Accuracy DM    (b) Health, Highly-Biased DM    (c) Health, Inconsistent DM

Figure 6: Comparing learning-to-defer, rejection learning and binary models. Health dataset. Each column is with a different DM scenario, according to captions. In left and centre columns, X-axis is fairness (lower is better); in right column, X-axis is deferral rate. Y-axis is accuracy for all. The red square is a baseline binary classifier, trained only to optimize accuracy; dashed line is the fair rejection model; solid line is a fair deferring model. Yellow circle shows DM alone. In left and centre columns, dotted line is a binary model also optimizing fairness. Each experiment is a hyperparameter sweep over $\gamma_{reject}/\gamma_{defer}$ (in left and centre columns, also $\alpha_{fair}$; in right column, $\alpha_{fair} = 0$; for results with $\alpha_{fair} \geq 0$, see Appendix G).

Results for Health dataset corresponding to Fig. 6 in Sec. 5.1.

# B    Results: Binary Classification with Fair Regularization

The results in Figures 7 and 8 roughly replicate the results from [2], who also test on the COMPAS dataset. Their results are slightly different for two reasons: 1) we use a 1-layer NN and they use logistic regression; and 2) our training/test splits are different from theirs - we have more examples in our training set. However, the main takeaway is similar: regularization is an effective way to reduce DI without making too many more errors. We also show the results of this regularization for neural networks with Bayesian uncertainty on the weights (see Appendix F).

(a) COMPAS, MLP    (b) Health, MLP    (c) COMPAS, BNN    (d) Health, BNN

Figure 7: Relationship of DI to $\alpha$, the coefficient on the DI regularizer, 5 runs for each value of $\alpha$. Two datasets, COMPAS and Health. Two learning algorithms, MLP and Bayesian weight uncertainty.

(a) COMPAS, MLP    (b) Health, MLP    (c) COMPAS, BNN    (d) Health, BNN

Figure 8: Relationship of error rate to $\alpha$, the coefficient on the DI regularizer, 5 runs for each value of $\alpha$. Two datasets, COMPAS and Health. Two learning algorithms, MLP and Bayesian weight uncertainty.

## C  Dataset and Experiment Details

We show results on two datasets. The first is the COMPAS recidivism dataset, made available by ProPublica [26] [2]. This dataset concerns recidivism: whether or not a criminal defendant will commit a crime while on bail. The goal is to predict whether or not the person will recidivate, and the sensitive variable is race (split into black and non-black). We used information about counts of prior charges, charge degree, sex, age, and charge type (e.g., robbery, drug possession). We provide one extra bit of information to our DM - whether or not the defendant *violently* recidivated. This clearly delineates between two groups in the data - one where the DM knows the correct answer (those who violently recidivated) and one where the DM has no extra information (those who did not recidivate, and those who recidivated non-violently). This simulates a real-world scenario where a DM, unbeknownst to the model, may have extra information on a subset of the data. The simulated DM had a 24% error rate, better than the baseline model's 29% error rate. We split the dataset into 7718 training examples and 3309 test examples.

The second dataset is the Heritage Health dataset[3]. This dataset concerns health and hospitalization, particularly with respect to insurance. For this dataset, we chose the goal of predicting the Charlson Index, a comorbidity indicator, related to someone's chances of death in the next several years. We binarize the Charlson Index of a patient as 0/greater than 0. We take the sensitive variable to be age and binarize by over/under 70 years old. This dataset contains information on sex, age, lab test, prescription, and claim details. The extra information available to the DM is the primary condition group of the patient (given in the form of a code e.g., 'SEIZURE', 'STROKE', 'PNEUM'). Again, this simulates the situation where a DM may have extra information on the patient's health that the algorithm does not have access to. The simulated DM had a 16% error rate, better than the baseline model's 21% error rate. We split the dataset into 46769 training examples and 20044 test examples.

We trained all models using a fully-connected two-layer neural network with a logistic non-linearity on the output, where appropriate. We used 5 sigmoid hidden units for COMPAS and 20 sigmoid hidden units for Health. We used ADAM [25] for gradient descent. We split the training data into 80% training, 20% validation, and stopped training after 50 consecutive epochs without achieving a new minimum loss on the validation set.

## D  Details on Optimization: Hard Thresholds

We now explain the post-hoc threshold optimization search procedure we used. To encourage fairness, we can learn a separate set of thresholds for each group, then apply the appropriate set of thresholds to each example. Since it is a very small space (one dimension per threshold = 4 dimensions; ), we used a random search. We sampled 1000 combinations of thresholds, picked the thresholds which minimized the loss on one half of the test set, and evaluated these thresholds on the other half of the test set. We do this for several values of $\alpha, \gamma$ in thresholding, as well as several values of $\alpha$ for the original binary model.

We did not sample thresholds from the [0, 1] interval uniformly. Rather we used the following procedure. We sampled our lower thresholds from the scores in the training set which were below 0.5, and our upper thresholds from the scores in the training set which were above 0.5. Our sampling scheme was guided by two principles: this forced 0.5 to always be in the PASS region; and this allowed us to sample more thresholds where the scores were more dense. If only choosing one threshold per class, we sampled from the entire training set distribution, without dividing into above 0.5 and below 0.5.

This random search was significantly faster than grid search, and no less effective. It was also faster and more effective than gradient-based optimization methods for thresholds - the loss landscape seemed to have many local minima.

|  |  |
|---|---|
| (a) COMPAS dataset | (b) Health dataset |

Figure 9: Comparing model performance between learning to defer training with oracle as DM to rejection learning. At test time, same DM is used.

## E    Comparison of Learning to Defer with an Oracle in Training to Rejection Learning

In Section 2.3, we discuss that rejection learning is similar to learning to defer training, except with a training DM who treats all examples similarly, in some sense. In Section 2.3, we show that theoretically these are equivalent. However, our fairness regularizer is not of the correct form for the proof in Sec.2.3 to hold. Here we show experimental evidence that the objectives are equivalent for the fair-regularized loss function. The plots in Figure 9 compare these two models: rejection learning, and learning to defer with an oracle at training time, and the standard DM at test time. We can see that these models trade off between accuracy and fairness in almost an identical manner.

## F    Bayesian Weight Uncertainty

We can also take a Bayesian approach to uncertainty by learning a distribution over the weights of a neural network [3]. In this method, we use variational inference to approximate the posterior distribution of the model weights given the data. When sampling from this distribution, we can obtain an uncertainty estimate. If sampling several times yields widely varying results, we can state the model is uncertain on that example.

This model outputs a prediction $p \in [0, 1]$ and an uncertainty $\rho \in [0, 1]$ for example $x$. We calculate these by sampling $J$ times from the model, yielding $J$ predictions $z_j \in [0, 1]$. Our prediction $p$ is the sample mean $\mu = \frac{1}{J} \sum_{j=1}^{J} z_j$. To represent our uncertainty, we can use signal-to-noise ratio, defined as $S = \frac{|\mu - 0.5|}{\sigma}$, based on $\mu$ and the sample standard deviation $\sigma = \sqrt{\frac{\sum_{j=1}^{J}(z_j - \mu)^2}{J-1}}$. Setting $\rho = \sigma(\log(1/S))$ yields uncertainty values in a $[0, 1]$ range. At test time, the system can threshold this uncertainty; any example with uncertainty beyond a threshold is rejected, and passed to the DM.

With weights $w$ and variational parameters $\theta$, our variational lower bound $\ell_m$ is then (with $CE$ denoting cross entropy):

$$\ell_m^{BNN}(Y, A, X, w; \theta) = -KL[q(w|\theta)||Prior(w)] + \mathbb{E}_{q(w|\theta)} - CE(Y_i, p(x_i; \theta))$$

## G    Results: Differentiable Learning-to-Defer Fairness with $\alpha_{fair} \geq 0$

We show here the results of the experiments with deferring fairly to a low accuracy, inconsistent DM with accuracy to extra information. The results are qualitatively similar to those in Figs. 3a and 6a. However, it is worth noting here that the rejection learning results mostly overlap the binary model results. This means that if the DM is not taken into consideration through learning to defer, then the win of rejection learning over training a binary model can be minimal.

(a) COMPAS dataset                              (b) Health dataset

Figure 10: Comparing learning-to-defer, rejection learning and binary models. High-accuracy, ignores fairness DM. X-axis is fairness (lower is better). Y-axis is accuracy. The red square is a baseline binary classifier, trained only to optimize accuracy; dashed line is the fair rejection model; solid line is a fair deferring model. Yellow circle shows DM alone. In left and centre columns, dotted line is a binary model also optimizing fairness. Each experiment is a hyperparameter sweep over $\gamma_{reject}/\gamma_{defer}/\alpha_{fair}$.

# H    Results: Learning to Defer by Deferral Rate

(a) COMPAS dataset, 0-30% De- (b) COMPAS dataset, 30-70% De- (c) COMPAS dataset, 70-100% ferral Rate                        ferral Rate                   Deferral Rate

Figure 11: Comparison of learning to defer (solid) and reject (dashed). Each figure considers only runs where the final deferral rate was low, medium, or high, taking the Pareto front on that set of examples.

Models which rarely defer behave very differently from those which frequently defer. In Figure 11, we break down the results from Figure 3a by deferral rate. First, we note that even for models with similar deferral rates, we see a similar fairness/accuracy win for the deferring models. Next, we can look separately at the low and high deferral rate models. We note that the benefit of learning to defer is much larger for high deferral rate models. This suggests that the largest benefit of learning to defer comes from a win in fairness, rather than accuracy, since the DM already provides high accuracy.

# I    Results: Deferral Rates with a Biased DM

In Fig. 12, we further analyze the difference in deferral and rejection rates between models trained with the biased DM (Fig. 3b) and standard DM (Fig. 3a). We ran the model for over 1000 different hyperparameter combinations, and show the distribution of the deferral rates of these runs on the COMPAS dataset, dividing up by defer/reject models, biased/standard DM, and the value of the sensitive attribute.

Notably, the deferring models are able to treat the two DM's differently, whereas the rejecting models are not. In particular, notice that the solid line (biased DM) is much higher in the low deferral regime, meaning that the deferring model, given a biased DM, almost defers on fewer than 20% of examples

(a) Reject, A = 0      (b) Defer, A = 0      (c) Reject, A = 1      (d) Defer, A = 1

Figure 12: Deferral rate for a range of hyperparameter settings, COMPAS dataset. X-axis is cutoff $\in [0, 1]$, line shows percentage of runs which had deferral rate below the cutoff. Blue solid line is models trained with biased DM, purple dashed line is with standard DM. Left column is rejection learning, right column is learning-to-defer. Top and bottom row split by value of the sensitive attribute $A$.

since the DM is of lower quality. Secondly, we see that the deferring model is also able to adapt to the biased DM by deferring at differing rates for the two values of the sensitive attribute — an effective response to a (mildly or strongly) biased DM who may already treat the two groups differently. This is another way in which a model that learns to defer can be more flexible and provide value to a system.

## Footnotes

[2]downloaded from https://github.com/propublica/compas-analysis

[3]Downloaded from https://www.kaggle.com/c/hhp