[Reviews · NeurIPS 2018]

Reviewer 1



Paper considers fairness in the context of machine learning systems that can refuse to make a prediction; and pass difficult instances to humans for them to label instead. Fairness in this new context is an interesting problem that requires understanding the biases and limitations of the human working with the classifier. The authors make a good case for their new approach being a useful generalisation of the existing learning to reject methods; that considers the accuracy and bias of the human response on particular instances. I like this idea, and I think it may become an important part of HCI in the future. The derivations are straightforward and flow appropriately from the definitions; and clearly show learning to reject as a special case of this approach. Experiments are understandably synthetic, using ML algorithms with additional information as a replacement for humans , but clearly make the case for the approach. In conclusion, this ticks all the boxes for a nips paper for me. Good, relevant idea, clear to follow, and sufficient synthetic experimental support. Using actual people in the experiments would make it even better, but this would introduce a whole host of new challenges and by no means is it necessary to show the merits of the approach. Minor errors: Typo cam-> can in abstract and wrong citation style used in paper.

Reviewer 2



The paper studies the interaction between an automated model and a decision-maker. The authors refer to this new model of learning as learning-to-defer and provide connections between this model, rejection learning and mixture of experts model of learning. The paper proposes approaches to solve learning-to-defer and experimentally evaluate these approaches on fairness-sensitive datasets. Questions: (1) Is there any particular reason for choosing neural networks as the binary classifier versus other methods? The COMPAS dataset has been studied for fairness-accuracy trade-offs in many other previous works with different learning methods. (2) What are the reasons behind picking the specific features as DM's extra information in the experiments? How do these features correlate with the label i.e. how informative these extra informations are? (3) Would the DM-fair (the binary model with DM's classifier that optimizes for fairness) trade-off curve dominate all the other curves in Figures 3.1/3.2? (4) Why is the deferral curve for the rejecting learning different than the deferral curve for learning-to-defer? In particular while the deferral curve for learning-to-defer is intuitive (there is an optimal deferring rate that results in the best accuracy and the accuracy degrades as we move further from that rate in each direction) the deferral curve for rejecting learning is surprisingly monotone! Strengths: (1) The paper is well-written and it is easy to follow/verify the main ideas in the first 4 sections of the paper (see my questions about the experimental section above and also regarding weaknesses below). (2) The generalization of rejection learning and connection to the literature on mixture of experts is neat and can be of independent interest. (3) The experimental evidence on how learning to defer adapts to DM's weakest subgroup is interesting and should be emphasized. Weaknesses: (1) The contributions of the paper would be more significant (or easier to quantify) if the authors compare their approach with many other related work that study the trade-off between fairness and accuracy with respect to equality of odds (e.g. Hardt et al 2016, Zafar et al 2017). Without such a comparison, it's hard to assess the significance of this paper in comparison to the growing literature on fairness in machine learning (see e.g. my question (4) above). Typos: (1) cam --> can in line 10. (2) extra space before the parenthesis in line 235. -------------------------------------------------------------- I read the other reviews and the rebuttal. The authors have convinced me that these direct comparisons might be difficult given the nature of their model. I like this paper as it explores novel and natural ideas. I think the paper is still only marginally above the acceptance threshold in its current form. I really hope that the authors clarify the presentation especially in the experimental section if the paper gets accepted.

Reviewer 3



Rebuttal: Thanks for the rebuttal. I appreciate the scenarios you detailed about my concerns of limited applicability due to "overfitting" to a DM. I did not give detecting common trends as well as crowd sourcing enough thought during reading (maybe make these examples more explicit) and agree that they appear quite practical. Given that I liked the approach to begin with, I changed my score to a 6. This paper formulates a two-stage decision making process, where one decision maker (an automated model) can either make the decision right away, or defer it to a second decision maker (typically human and with access to additional information, but potentially biased). The main extension beyond rejection learning is that the automated model has access and adapts to the (human) decision maker's output during training, which allows not only to exploit the decision makers strengths (additional information), but also make up for its weaknesses (bias against subgroups, uncertainty on certain subgroups), in applications for fair decision making. The paper is clearly structured, well written and very well motivated. Except for minor confusions about some of the math, I could easily follow and enjoyed reading the paper. As far as I know, the framework and particularly the application to fairness is novel. I believe the general idea of incorporating and adjusting to human decision makers as first class citizens of the pipeline is important for the advancement of fairness in machine learning. However, the framework still seems to encompass a rather minimal technical contribution in the sense that both a strong theoretical analysis and exhaustive empirical evaluation are lacking. Moreover, I am concerned about the real world applicability of the approach, as it mostly seems to concern situations with a rather specific (but unknown) behavior of the decision maker, which typically does not transfer across DMs, needs to be known during training. I have trouble thinking of situations where sufficient training data, both ground truth and the DMs predictions, are available simultaneously. While the authors do a good job evaluating various aspects of their method (one question about this in the detailed comments), those are only two rather simplistic synthetic scenarios. Because of the limited technical and experimental contribution, I heavy-heartedly tend to vote for rejection of the submission, even though I am a big fan of the motivation and approach. Detailed Comments - I like the setup description in Section 2.1. It is easy to follow and clearly describes the technical idea of the paper. - I have trouble understanding (the proof of) the Theorem (following line 104). You show that eq (6) and eq (7) are equal for appropriately chosen $\gamma_{defer}$. However, (7) is not the original deferring loss from eq (3). Shouldn't the result be that learning to defer and rejection learning are equivalent if for the (assumed to be) constant DM loss, $\alpha$ happens to be equal to $\gamma_{reject}$? In the theorem it sounds as if they were equivalent independent of the parameter choices for $\gamma_{reject}$ and $\alpha$. The main takeaway, namely that there is a one-to-one correspondence between rejection learning with cost $\gamma_{reject}$ and learning to defer with a DM with constant loss $\alpha$, is still true. Is there a specific reason why the authors decided to present the theorem and proof in this way? - The authors highlight various practical scenarios in which learning to defer is preferable and detail how it is expected to behave. However, this practicability seems to be heavily impaired by the strong assumptions necessary to train such model, i.e., availability of ground truth and DM's decisions for each DM of interest, where each is expected to have their own specific biases/uncertainties/behaviors during training. - What does it mean for the predictions \hat{Y} to follow an (independent?) Bernoulli equation (12) and line 197? How is p chosen, and where does it enter? Could you improve clarity by explicitly stating w.r.t. what the expectations in the first line in (12) are taken (i.e., where does p enter explicitly?) Shouldn't the expectation be over the distribution of \hat{Y} induced by the (training) distribution over X? - In line 210: The impossibility results only hold for (arguably) non-trivial scenarios. - When predicting the Charlson Index, why does it make sense to treat age as a sensitive attribute? Isn't age a strong and "fair" indicator in this scenario? Or is this merely for illustration of the method? - In scenario 2 (line 252), does $\alpha_{fair}$ refer to the one in eq (11)? Eq. (11) is the joint objective for learning the model (prediction and deferral) given a fixed DM? That would mean that the autodmated model is encouraged to provide unfair predictions. However, my intuition for this scenario is that the (blackbox) DM provides unfair decisions and the model's task is to correct for it. I understand that the (later fixed) DM is first also trained (semi synthetic approach). Supposedly, unfairness is encouraged only when training DM as a pre-stage to learning the model? I encourage the authors to draw the distinction between first training/simulating the DM (and the corresponding assumptions/parameters) and then training the model (and the corresponding assumptions/parameters) more clearly. - The comparison between the deferring and the rejecting model is not quite fair. The rejecting model receives a fixed cost for rejecting and thus does not need access to DM during training. This already highlights that it cannot exploit specific aspects (e.g., additional information) of the DM. On the other hand, while the deferring model can adaptively pass on those examples to DM, on which the DM performs better, this requires access to DM's predictions during training. Since DMs typically have unique/special characteristics that could vary greatly from one DM to the next, this seems to be a strong impairment for training a deferring model (for each DM individually) in practice? While the adaptivity of learning to defer unsurprisingly constitutes an advantage over rejection learning, it comes at the (potentially large) cost of relying on more data. Hence, instead of simply showing its superiority over rejection learning, one should perhaps evaluate this tradeoff? - Nitpicking: I find "above/below diagonal" (add a thin gray diagonal to the plot) easier to interpret than "above/below 45 degree", which sounds like a local property (e.g., not the case where the red line saturates and has "0 degrees"). - Is the slight trend of the rejecting model on the COMPAS dataset in Figure 4 to defer less on the reliable group a property of the dataset? Since rejection learning is non-adaptive, it is blind to the properties of DM, i.e., one would expect it to defer equally on both groups if there is no bias in the data (greater variance in outcomes for different groups, or class imbalance resulting in higher uncertainty for one group). - In lines 306-307 the authors argue that deferring classifiers have higher overall accuracy at a given minimum subgroup accuracy (MSA). Does that mean that at the same error rate for the subgroup with the largest error rate (minimum accuracy), the error rate on the other subgroups is on average smaller (higher overall accuracy)? This would mean that the differences in error rates between subgroups are larger for the deferring classifier, i.e., less evenly distributed, which would mean that the deferring classifier is less fair? - Please update the references to point to the conference/journal versions of the papers (instead of arxiv versions) where applicable. Typos line 10: learning to defer ca*n* make systems... line 97: first "the" should be removed End of line 5 of the caption of Figure 3: Fig. 3a (instead of Figs. 3a) line 356: This reference seems incomplete?